# Analytic solutions of variance swaps for Heston models with stochastic long-run mean of variance and jumps

**Jing Fu** *

School of Mathematics, Southwestern University of Finance and Economics, Chengdu, China

* 1200202z1009@smail.swufe.edu.cn

## Abstract

This paper presents the pricing formulas for variance swaps within the Heston model that incorporates jumps and a stochastic long-term mean for the underlying asset. By leveraging the Feynman-Kac theorem, we derive a partial integro-differential equation (PIDE) to obtain the joint moment-generating function for the aforementioned model. Furthermore, we provide a series pricing formula for discretely sampled variance swap, derived through the use of this joint moment-generating function. Additionally, we discuss the limiting properties of the pricing formula for discretely sampled variance swap, namely, the pricing formula for continuously sampled variance swap. Finally, to demonstrate the efficacy of the pricing formula, we conduct several numerical simulation experiments, including comparisons with Monte Carlo (MC) simulation results and an analysis of the impact of parameter variations on the strike price of variance swaps.

## 1 Introduction

Currently, as financial markets become increasingly complex, volatility risk has garnered significant attention in the financial industry. In order to effectively address this risk, volatility derivatives have become essential tools for measuring and managing financial risk. With the sharp increase in trading volumes, the precise and efficient determination of volatility derivative prices has emerged as a central topic in quantitative finance and risk management.

In 1973, the classic option pricing model proposed by Black and Scholes [1] and Merton [2] assumed constant asset price volatility. However, empirical studies have shown that the implied volatility reverse-engineered from the Black-Scholes formula is often non-constant, exhibiting the "volatility smile" phenomenon. This suggests that stock price volatility may be related to its level, time, or external random factors. As a result, many researchers have turned to studying stochastic volatility models, with the Stochastic Volatility (SV) model receiving particular attention.

Subsequently, Hull and White [3] introduced a continuous-time stochastic volatility model and derived a European option pricing formula via a second-order Taylor expansion. Around the same time, Scott [4] and Stein [5] presented modeling volatility using an Ornstein-Uhlenbeck process with mean-reverting characteristics and provided closed-form pricing formulas. However, their model could not prevent volatility from becoming negative, and

**Data availability statement:** All relevant data are within the manuscript and its Supporting information files. Additionally, data has been

deposited on FigShare and can be retrieved via the following DOI: 10.6084/m9.figshare.28194263.

**Funding:** The author(s) received no specific funding for this work.

**Competing interests:** The authors have declared that no competing interests exist.

the assumption of zero correlation between the underlying asset price and volatility was somewhat unrealistic. A few years later, Schöbel and Zhu [6] extended a model allowing for correlation between the underlying stock returns and instantaneous volatility, obtaining a closed-form solution for the European option pricing formula through an inverse Fourier transform method.

Furthermore, Heston [7] modeled volatility using the CIR model and derived a closed-form solution for the standard European option pricing through affine structure and Fourier transform methods. This affine stochastic volatility model not only provides a closed-form pricing formula but also adeptly captures the phenomenon of the "volatility smile", making it a fundamental model for pricing volatility derivatives and instigating widespread research efforts. For instance, Elliott and Lian [8] studied the pricing of variance and volatility swaps under discrete observations using a segmented Heston stochastic volatility model, obtaining accurate closed-form solutions. He and Zhu [9] developed a hybrid model combining CIR random interest rates and Heston stochastic volatility to analyze the pricing formula for variance and volatility swaps, demonstrating convergence and verifying the accuracy of swap prices. Additionally, Kim and Kim [10] addressed the pricing issue of generalized variance swaps within the Heston-CIR hybrid model, offering exact solutions for the fair strike prices of these swaps. He and Lin [11] introduced a three-factor model incorporating random volatility and interest rates, deriving an exact analytical pricing formula for foreign exchange options. Najafi and Mehrdoust [12] analyzed the pricing of European options with proportional transaction costs in an incomplete market setting using two long-memory versions of the Heston model. However, the Heston model does have certain limitations, for example, the square root specification is typically not suitable for modeling index returns [13,14]. Additionally, Bakshi et al. [15] pointed out that the volatility process often exhibits non-linear mean-reverting characteristics.

Considering that the constant long-term average variance in the Heston model fails to capture the temporal market volatility dynamics, Byelkina and Levin [16] and Forde and Jacquier [17] suggested introducing a time-dependent mean-reverting variance level to better capture the term structure of implied volatility and variance swap curves. He and Chen [18] proposed a novel stochastic volatility model where the long-term mean of volatility in the Heston model itself follows a stochastic process, deriving a closed-form pricing formula for European options within this framework. Their empirical study, employing adaptive sample annealing, demonstrated the superiority of this pricing model over the Heston model. Yoon et al. [19] further extended the long-term variance mean of the Heston model to a stochastic process with mean-reverting characteristics and derived fair strike prices for variance swaps under this setup. Experimental results indicate that the stochastic long-run mean of variance plays a significant role in determining the fair strike prices of variance swaps in turbulent market.

Meanwhile, financial markets are often subject to sharp price fluctuations due to external shocks, such as major political events, economic data releases, or natural disasters, which continuous asset pricing models may not fully capture. To better represent asset price dynamics, researchers have incorporated jump risks into continuous models. For instance, Zheng and Kwok [20] derived pricing formulas for generalized variance swaps within a stochastic volatility framework where both the asset price and variance process allow for simultaneous jumps. Cui et al. [21] utilized frame duality and density projection method, combined with a novel weak approximation scheme based on continuous-time Markov chain (CTMC), to provide pricing formulas for volatility derivatives under discrete sampling. Wu et al. [22] proposed a variance swap valuation model that combines multi-factor stochastic spot variance and long-term variance, while allowing for mean reversion in asset prices and a co-jump structure. Empirical results demonstrate that this model outperforms other models. Wang and Guo [23]

analyzed the pricing of variance and volatility swaps within the framework of a double Heston jump-diffusion model with approximate fractional stochastic volatility. Additionally, Ascione et al. [24] combined the Heston-CIR model with a Lévy process to enhance pricing accuracy in the foreign exchange (FX) market, presenting a new formula that more closely aligns with the observed price distribution.

The main goal of this paper is to propose a novel stochastic volatility model that includes jump processes and a stochastic long-term mean for variance, and to investigate the pricing of variance swaps within this framework. This model, which integrates asset price jumps and a stochastic long-term mean for variance, is more comprehensive than existing models.

The rest of this paper is organized as follows: Sect 2 introduces the new model and obtains its joint moment generating function (MGF) in a power series form. Sect 3 utilizes the joint MGF derived in Sect 2 to develop pricing formulas for variance swaps under both discrete and continuous sampling. Sect 4 show cases numerical experiments and examples to demonstrate the importance of incorporating a stochastic long-term mean. The concluding section offers final remarks.

## 2 The newly proposed model and its MGF

In this section, we aim to propose a new model based on the Yoon et al. [19]'s model , and derive the joint moment generating function(MGF) for this newly models. Consider the risk-neutral probability measurable space $(\Omega, \mathcal{F}_t, \mathbb{Q})$ in which we give the newly models as follows

$$\begin{cases} \frac{dS_t}{S_t} = (r - d - \lambda m)dt + \sqrt{v_t}dW_t^S + (e^{J^S} - 1)dN_t, \\ dv_t = \kappa_v(\theta_t - v_t)dt + \sigma_v\sqrt{v_t}dW_t^v + J^v dN_t, \\ d\theta_t = \kappa_\theta(\tilde{\theta} - \theta_t)dt + \sigma_\theta dW_t^\theta, \end{cases} \tag{1}$$

where $r, d, \kappa_v, \kappa_\theta, \tilde{\theta}, \sigma_v$ and $\sigma_\theta$ are positive constants, $\lambda$ denotes the jump intensity and $m = \mathbb{E}^{\mathbb{Q}}[e^{J^S} - 1]$ represents the average jump amplitude of the price. Moreover, $W_t^S, W_t^v$ and $W_t^\theta$ are standard Brownian motions with $< dW_t^S, dW_t^v >= \rho_{sv}dt$ and $< dW_t^S, dW_t^\theta >=< dW_t^v, dW_t^\theta >= 0$, $N_t$ is independent with $W_t^S$ and $W_t^v$. $J^S$ and $J^v$ represent the jump sizes of the price and variance, respectively, assumed to be independent of $W_t^S, W_t^v$, and $N_t$. For easy of solving the analytic solutions, we further assume that

$$J^v \sim \exp\left(\frac{1}{\eta}\right), \qquad J^S \mid J^v \sim \mathrm{N}\left(v + \rho_J J^v, \delta^2\right).$$

Note that the proposed new model (1) encompasses several well-known models as special cases. The first is that $\theta_t = \theta$ a constant, it is reduced to the case considered in Zheng and Kwok [20]. The second is that no jump diffusion occurs in model (1) , it is reduced to the case considered in Yoon et al. [19]. The third is that both $\theta_t$ a constant and no jump diffusion occurs in model (1) , it is reduced to the classic Heston stochastic volatility model. For convenience, we define $x_t = \ln S_t$. To achieve the analytic solutions for variance swaps to the new model (1), we end this section by providing the expression for the joint moment generating function (MGF) of the joint process $x_t, v_t$, and $\theta_t$ shown as follows.

**2.1.** *Assume that the dynamics of the asset price $S_t$ follows dynamics Eq (1) and denote*

$$U(x_t, v_t, \theta_t, t) = \mathbb{E}^{\mathbb{Q}}\left[e^{\phi x_T + bv_T + c\theta_T + \gamma}\big|\mathcal{F}_t\right],$$

*Then we have*

$$U\left(x_t, v_t, \theta_t, t\right) = \mathrm{e}^{\phi x_t + B(q,\tau)v_t + C(q,\tau)\theta_t + D(q,\tau) + E(q,\tau)}, \tag{2}$$

*where $\tau = T - t$ and*

$$B(q,\tau) = \frac{1}{\sigma_v^2}\left[\left(\kappa_v - \rho_{sv}\sigma_v\phi\right) - \delta_1 \cdot \frac{\sinh\left(\frac{\delta_1}{2}\tau\right) + \delta_2\cosh\left(\frac{\delta_1}{2}\tau\right)}{\cosh\left(\frac{\delta_1}{2}\tau\right) + \delta_2\sinh\left(\frac{\delta_1}{2}\tau\right)}\right],$$

$$C(q,\tau) = -\frac{\kappa_v\delta_1\mathrm{e}^{-\kappa_\theta\tau}}{\sigma_v^2}\left[\sum_{n=0}^{\infty}(-u)^n\left(\frac{\mathrm{e}^{(\kappa_\theta - n\delta_1)\tau} - 1}{\kappa_\theta - n\delta_1} - u\frac{\left(\mathrm{e}^{(\kappa_\theta - (n+1)\delta_1)\tau} - 1\right)}{\kappa_\theta - (n+1)\delta_1}\right)\right]$$
$$+ \frac{\kappa_v\left(\kappa_v - \rho_{sv}\sigma_v\phi\right)}{\kappa_\theta\sigma_v^2}\left(1 - \mathrm{e}^{-\kappa_\theta\tau}\right) + c\mathrm{e}^{-\kappa_\theta\tau},$$

$$D(q,\tau) = (r - d)\phi\tau + \frac{\tilde{\theta}\kappa_v}{\sigma_v^2}\left[\left(\kappa_v - \rho_{sv}\sigma_v\phi\right)\tau - 2\ln\left(\cosh\left(\frac{\delta_1}{2}\tau\right) + \delta_2\sinh\left(\frac{\delta_1}{2}\tau\right)\right)\right]$$
$$+ \frac{\kappa_v^2\sigma_\theta^2\left(\kappa_v - \rho_{sv}\sigma_v\phi\right)}{2\kappa_\theta^2\sigma_v^4}\left[\left(\kappa_v - \rho_{sv}\sigma_v\phi\right)\tau - 2\ln\left(\cosh\left(\frac{\delta_1}{2}\tau\right) + \delta_2\sinh\left(\frac{\delta_1}{2}\tau\right)\right)\right]$$
$$- \left(\frac{\kappa_v\sigma_\theta^2\left(\kappa_v - \rho_{sv}\sigma_v\phi\right)}{2\kappa_\theta^2\sigma_v^2} + \tilde{\theta}\right)\left(C(q,\tau) - c\right) - \frac{\kappa_v\sigma_\theta^2}{4\kappa_\theta}\left(C(q,\tau)^2 - c^2\right)$$
$$+ \frac{\kappa_v^2\sigma_\theta^2\delta_1\left(\kappa_v - \rho_{sv}\sigma_v\phi\right)}{2\kappa_\theta^2\sigma_v^4}\left[-\frac{2}{\delta_1}\ln\left(\cosh\left(\frac{\delta_1}{2}\tau\right) + \delta_2\sinh\left(\frac{\delta_1}{2}\tau\right)\right)\right.$$
$$\left. + \sum_{i=0}^{\infty}(-u)^i\left(\frac{\mathrm{e}^{-(\kappa_\theta + i\delta_1)\tau} - 1}{-(\kappa_\theta + i\delta_1)} - u\cdot\frac{\mathrm{e}^{-(\kappa_\theta + (i+1)\delta_1)\tau} - 1}{-(\kappa_\theta + (i+1)\delta_1)}\right)\right]$$
$$- \frac{\kappa_v\sigma_\theta^2\delta_1 c}{2\kappa_\theta\sigma_v^2}\left[\sum_{i=0}^{\infty}(-u)^i\left(\frac{\mathrm{e}^{-(\kappa_\theta + i\delta_1)} - 1}{-(\kappa_\theta + i\delta_1)} - u\frac{\mathrm{e}^{-(\kappa_\theta + (i+1)\delta_1)\tau} - 1}{-(\kappa_\theta + (i+1)\delta_1)}\right)\right]$$
$$+ \frac{\kappa_v^2\delta_1^2\sigma_\theta^2}{2\kappa_\theta\sigma_v^4}\left[\frac{\tau}{\kappa_\theta} + \frac{\mathrm{e}^{-\kappa_\theta\tau} - 1}{\kappa_\theta^2} + \sum_{i=1}^{\infty}\frac{(-u)^i}{\kappa_\theta - i\delta_1}\left(\frac{\mathrm{e}^{-i\delta_1\tau} - 1}{-i\delta_1} - \frac{\mathrm{e}^{-\kappa_\theta\delta_1\tau} - 1}{-\kappa_\theta}\right)\right.$$
$$+ \sum_{i=0}^{\infty}\frac{(-u)^{i+1}}{\kappa_\theta - (i+1)\delta_1}\left(\frac{\mathrm{e}^{-(i+1)\delta_1\tau} - 1}{-(i+1)\delta_1} - \frac{\mathrm{e}^{-\kappa_\theta\delta_1\tau} - 1}{-\kappa_\theta}\right)$$
$$+ \sum_{n=1}^{\infty}\sum_{i=0}^{\infty}\frac{(-u)^{n+i}}{\kappa_\theta - i\delta_1}\left(\frac{\mathrm{e}^{-(n+i)\delta_1\tau} - 1}{-(n+i)\delta_1} - \frac{\mathrm{e}^{-(\kappa_\theta + n\delta_1)\tau} - 1}{-(\kappa_\theta + n\delta_1)}\right)$$
$$+ \sum_{n=1}^{\infty}\sum_{i=0}^{\infty}\frac{(-u)^{n+i+1}}{\kappa_\theta - (i+1)\delta_1}\left(\frac{\mathrm{e}^{-(n+i+1)\delta_1\tau} - 1}{-(n+i+1)\delta_1} - \frac{\mathrm{e}^{-(\kappa_\theta + n\delta_1)\tau} - 1}{-(\kappa_\theta + n\delta_1)}\right)$$
$$+ \sum_{n=0}^{\infty}\sum_{i=0}^{\infty}\frac{(-u)^{n+i+1}}{\kappa_\theta - i\delta_1}\left(\frac{\mathrm{e}^{-(n+i+1)\delta_1\tau} - 1}{-(n+i+1)\delta_1} - \frac{\mathrm{e}^{-(\kappa_\theta + (n+1)\delta_1)\tau} - 1}{-(\kappa_\theta + (n+1)\delta_1)}\right)$$
$$\left. + \sum_{n=0}^{\infty}\sum_{i=0}^{\infty}\frac{(-u)^{n+i+2}}{\kappa_\theta - (i+1)\delta_1}\left(\frac{\mathrm{e}^{-(n+i+2)\delta_1\tau} - 1}{-(n+i+2)\delta_1} - \frac{\mathrm{e}^{-(\kappa_\theta + (n+1)\delta_1)\tau} - 1}{-(\kappa_\theta + (n+1)\delta_1)}\right)\right],$$

$$E(q,\tau) = -\lambda(m\phi + 1)\tau + \frac{\lambda\sigma_v^2\mathrm{e}^{\phi v + \frac{\phi^2\delta_1^2}{2}}}{m_1 + \eta\delta_1}\tau$$
$$- \lambda\sigma_v^2\mathrm{e}^{\phi v + \frac{\phi^2\delta^2}{2}}\frac{2\eta}{m_1^2 - \eta^2\delta_1^2}\ln\frac{m_1 + \eta\delta_1 + (m_1 u - \eta\delta_1 u)\,\mathrm{e}^{-\delta_1\tau}}{m_1 + \eta\delta_1 + (m_1 u - \eta\delta_1 u)},$$

with $q = (\phi, b, c, r)^T$ and

$$u = \frac{1 - \delta_2}{1 + \delta_2}, \qquad m_1 = \sigma_v^2 - \eta \left( \kappa_v - \rho_{sv}\sigma_v\phi + \sigma_v^2\phi\rho_J \right),$$

with

$$\delta_1 = \sqrt{\left(\kappa_v - \rho_{sv}\sigma_v\phi\right)^2 + \sigma_v^2\left(\phi - \phi^2\right)}, \qquad \delta_2 = \frac{\left(\kappa_v - \rho_{sv}\sigma_v\phi\right) - b\sigma_v^2}{\delta_1}.$$

*Proof*: By using Feynman-Kac theorem, we obtain $U(x_t, v_t, \theta_t, t)$ satisfies the following PIDE equation

$$
\begin{aligned}
\frac{\partial U}{\partial \tau} =& \left(r - d - \lambda m - \frac{v}{2}\right)\frac{\partial U}{\partial x} + \kappa_v(\theta - v)\frac{\partial U}{\partial v} + \kappa_\theta\left(\tilde{\theta} - \theta\right)\frac{\partial U}{\partial \theta} \\
&+ \frac{v}{2}\frac{\partial^2 U}{\partial x^2} + \frac{\sigma_v^2}{2}v\frac{\partial^2 U}{\partial v^2} + \frac{\sigma_\theta^2}{2}\frac{\partial^2 U}{\partial \theta^2} + \rho_{sv}\sigma_v v\frac{\partial^2 U}{\partial x\partial v} \\
&+ \lambda\mathbb{E}^{\mathbb{Q}}\left[U\left(x + J^S, v + J^v, \theta, \tau\right) - U(x, v, \theta, \tau)\right],
\end{aligned}
\tag{3}
$$

with $U|_{\tau=0} = e^{\phi x_T + b v_T + c\theta_T + \gamma}$. Since Eq (3) has affine structure, we guess that $U(x_t, v_t, \theta_t, t)$ admits an analytic solution with form given by

$$U(x_t, v_t, \theta_t, \tau) = e^{\phi x_t + B(q,\tau)v_t + C(q,\tau)\theta_t + D(q,\tau) + E(q,\tau)}. \tag{4}$$

By putting Eq (4) into Eq (3), it gives

$$
\begin{aligned}
v\frac{\partial B}{\partial \tau} + \theta\frac{\partial C}{\partial \tau} + \frac{\partial D}{\partial \tau} + \frac{\partial E}{\partial \tau} =& \left(r - d - \lambda m - \frac{v}{2}\right)\phi + \kappa_v(\theta - v)B + \kappa_\theta(\tilde{\theta} - \theta)C + \frac{v}{2}\phi^2 \\
&+ \frac{\sigma_v^2}{2}vB^2 + \frac{\sigma_\theta^2}{2}C^2 + \rho_{sv}\sigma_v v\phi B + \lambda\mathbb{E}^{\mathbb{Q}}\left[e^{\phi J^S + BJ^v} - 1\right],
\end{aligned}
$$

which by separating variables can be divided into four ODEs

$$
\begin{cases}
\frac{\partial B}{\partial \tau} = -\frac{1}{2}\left(\phi - \phi^2\right) - \left(\kappa_v - \rho_{sv}\sigma_v\phi\right)B + \frac{\sigma_v^2}{2}B^2, \\
\frac{\partial C}{\partial \tau} = \kappa_v B - \kappa_\theta C, \\
\frac{\partial D}{\partial \tau} = (r - d)\phi + \kappa_\theta\tilde{\theta}C + \frac{\sigma_\theta^2}{2}C^2, \\
\frac{\partial E}{\partial \tau} = \lambda\left(\mathbb{E}^{\mathbb{Q}}\left[e^{\phi J^S + BJ^v} - 1\right] - m\phi\right),
\end{cases}
\tag{5}
$$

with initial conditions

$$B(q, 0) = b, \quad C(q, 0) = c, \quad D(q, 0) = \gamma, \quad E(q, 0) = 0.$$

Obviously, the ODE governing $B(\phi, \tau)$ is a Riccati equation with constant coefficients, which is not difficult to solve. Thus, the details are omitted.

By integrating both sides of the ODE governing $C(\phi, \tau)$ and assuming

$$\frac{1}{1 + ue^{-\delta_1\tau}} = \sum_{n=0}^{\infty}(-u)^n e^{-n\delta_1\tau}, \quad |u| < 1.$$

The expression of $C(\phi, \tau)$ can be easily worked out, one has

$$C(q, \tau) = -\frac{\kappa_v \delta_1 e^{-\kappa_\theta \tau}}{\sigma_v^2} \left[ \sum_{n=0}^{\infty} (-u)^n \left( \frac{e^{(\kappa_\theta - n\delta_1)\tau} - 1}{\kappa_\theta - n\delta_1} - u \frac{\left(e^{(\kappa_\theta - (n+1)\delta_1)\tau} - 1\right)}{\kappa_\theta - (n+1)\delta_1} \right) \right]$$
$$+ \frac{\kappa_v (\kappa_v - \rho_{sv}\sigma_v\phi)}{\kappa_\theta \sigma_v^2} \left(1 - e^{-\kappa_\theta \tau}\right) + ce^{-\kappa_\theta \tau},$$

Furthermore,

$$D(q, \tau) = (r - d)\phi\tau + \kappa_\theta \tilde{\theta} \int_0^\tau C(q, s)ds + \int_0^\tau \frac{\sigma_\theta^2}{2} C^2(q, s)ds + \gamma. \tag{6}$$

By exploiting the ODEs Eq (5), we are able to obtain the relation

$$\begin{cases} \kappa_\theta \int_0^\tau C(q, s)ds = \kappa_v \int_0^\tau B(q, s)ds - C(q, \tau) + c, \\ \int_0^\tau C^2(q, s)ds = \frac{1}{\kappa_\theta} \left[ \kappa_v \int_0^\tau B(q, s)C(q, s)ds - \frac{1}{2}C^2(q, \tau) + \frac{1}{2}C^2 \right]. \end{cases} \tag{7}$$

Applying Eq (7) to Eq (6), $D(q, \tau)$ transforms into

$$D(q, \tau) = (r - d)\phi\tau + \tilde{\theta} \left[ \kappa_v \int_0^\tau B(q, s)ds - C(q, \tau) + c \right]$$
$$+ \frac{\sigma_\theta^2}{2\kappa_\theta} \left[ \kappa_v \int_0^\tau B(q, s)C(q, s)ds - \frac{1}{2}C^2(q, \tau) + \frac{1}{2}C^2 \right] + \gamma. \tag{8}$$

To calculate Eq (8), we need to integrate $B(q, s)$ and $B(q, s)C(q, s)$. Through direct calculation, the integral of $B(q, s)$ yields

$$\int_0^\tau B(q, s)ds = \frac{1}{\sigma_v^2} \left[ (\kappa_v - \rho_{sv}\sigma_v\phi)\tau - 2\ln\left(\cosh\left(\frac{\delta_1}{2}\tau\right) + \delta_2 \sinh\left(\frac{\delta_1}{2}\tau\right)\right) \right], \tag{9}$$

and the integral of $B(q, s)C(q, s)$ can be computed as

$$\frac{\kappa_v \sigma_\theta^2}{2\kappa_\theta} \int_0^\tau B(q, s)C(q, s)ds$$
$$= \frac{\kappa_v^2 \sigma_\theta^2 (\kappa_v - \rho_{sv}\sigma_v\phi)}{2\kappa_\theta^2 \sigma_v^4} \left[ (\kappa_v - \rho_{sv}\sigma_v\phi)\tau - 2\ln\left(\cosh\left(\frac{\delta_1}{2}\tau\right) + \delta_2 \sinh\left(\frac{\delta_1}{2}\tau\right)\right) \right]$$
$$- \left( \frac{\kappa_v \sigma_\theta^2 (\kappa_v - \rho_{sv}\sigma_v\phi)}{2\kappa_\theta^2 \sigma_v^2} \right) (C(q, \tau) - c)$$
$$+ \frac{\kappa_v^2 \sigma_\theta^2 \delta_1 (\kappa_v - \rho_{sv}\sigma_v\phi)}{2\kappa_\theta^2 \sigma_v^4} \left[ -\frac{2}{\delta_1} \ln\left(\cosh\left(\frac{\delta_1}{2}\tau\right) + \delta_2 \sinh\left(\frac{\delta_1}{2}\tau\right)\right) \right.$$
$$\left. + \sum_{i=0}^{\infty} (-u)^i \left( \frac{e^{-(\kappa_\theta + i\delta_1)\tau} - 1}{-(\kappa_\theta + i\delta_1)} - u \cdot \frac{e^{-(\kappa_\theta + (i+1)\delta_1)\tau} - 1}{-(\kappa_\theta + (i+1)\delta_1)} \right) \right]$$
$$- \frac{\kappa_v \sigma_\theta^2 \delta_1 c}{2\kappa_\theta \sigma_v^2} \left[ \sum_{i=0}^{\infty} (-u)^i \left( \frac{e^{-(\kappa_\theta + i\delta_1)} - 1}{-(\kappa_\theta + i\delta_1)} - u \frac{e^{-(\kappa_\theta + (i+1)\delta_1)\tau} - 1}{-(\kappa_\theta + (i+1)\delta_1)} \right) \right]$$
$$+ \frac{\kappa_v^2 \delta_1^2 \sigma_\theta^2}{2\kappa_\theta \sigma_v^4} \left[ \frac{\tau}{\kappa_\theta} + \frac{e^{-\kappa_\theta \tau} - 1}{\kappa_\theta^2} + \sum_{i=1}^{\infty} \frac{(-u)^i}{\kappa_\theta - i\delta_1} \left( \frac{e^{-i\delta_1 \tau} - 1}{-i\delta_1} - \frac{e^{-\kappa_\theta \delta_1 \tau} - 1}{-\kappa_\theta} \right) \right]$$

$$+ \sum_{i=0}^{\infty} \frac{(-u)^{i+1}}{\kappa_\theta - (i+1)\delta_1} \left( \frac{e^{-(i+1)\delta_1\tau} - 1}{-(i+1)\delta_1} - \frac{e^{-\kappa_\theta \delta_1 \tau} - 1}{-\kappa_\theta} \right)$$

$$+ \sum_{n=1}^{\infty} \sum_{i=0}^{\infty} \frac{(-u)^{n+i}}{\kappa_\theta - i\delta_1} \left( \frac{e^{-(n+i)\delta_1\tau} - 1}{-(n+i)\delta_1} - \frac{e^{-(\kappa_\theta + n\delta_1)\tau} - 1}{-(\kappa_\theta + n\delta_1)} \right)$$

$$+ \sum_{n=1}^{\infty} \sum_{i=0}^{\infty} \frac{(-u)^{n+i+1}}{\kappa_\theta - (i+1)\delta_1} \left( \frac{e^{-(n+i+1)\delta_1\tau} - 1}{-(n+i+1)\delta_1} - \frac{e^{-(\kappa_\theta + n\delta_1)\tau} - 1}{-(\kappa_\theta + n\delta_1)} \right)$$

$$+ \sum_{n=0}^{\infty} \sum_{i=0}^{\infty} \frac{(-u)^{n+i+1}}{\kappa_\theta - i\delta_1} \left( \frac{e^{-(n+i+1)\delta_1\tau} - 1}{-(n+i+1)\delta_1} - \frac{e^{-(\kappa_\theta + (n+1)\delta_1)\tau} - 1}{-(\kappa_\theta + (n+1)\delta_1)} \right)$$

$$+ \sum_{n=0}^{\infty} \sum_{i=0}^{\infty} \frac{(-u)^{n+i+2}}{\kappa_\theta - (i+1)\delta_1} \left( \frac{e^{-(n+i+2)\delta_1\tau} - 1}{-(n+i+2)\delta_1} - \frac{e^{-(\kappa_\theta + (n+1)\delta_1)\tau} - 1}{-(\kappa_\theta + (n+1)\delta_1)} \right) \Bigg]. \tag{10}$$

Substituting Eq (9) and Eq (10) into Eq (8) yields the expression for $D(q, \tau)$.

Due to $J^\nu \sim \exp\left(\frac{1}{\eta}\right), J^S \mid J^\nu \sim \mathrm{N}\left(\nu + \rho_J J^\nu, \delta^2\right)$, we have

$$\frac{\partial E}{\partial \tau} = \lambda E^{\mathbb{Q}} \left[ \left( e^{\phi J^S + B J^\nu} - 1 \right) \right] - \lambda m \phi$$

$$= -\lambda(m\phi + 1) + \lambda e^{\phi \nu + \frac{\delta^2 \phi^2}{2}} \frac{1}{1 - (\phi \rho_J B)\eta}. \tag{11}$$

Solving Eq (11) completes the proof. □

## 3 Analytic solutions for variance swaps

In this section, we introduce variance swaps and using the joint moment generating function to derive the pricing formulas for variance swaps under the frame of discrete and continuous sampling, respectively.

To do this, we first give the definitions of the annualized realized variance and volatility. According to Yuen et al. [25], the calculation of realized variance depends on whether it is based on the actual or log returns. The annualized realized variance of the actual returns and the log returns are represented as $RV_{var}^{(1)}$ and $RV_{var}^{(2)}$, respectively, which are given by

$$RV_{var}^{(1)} = \frac{AF}{N} \sum_{i=1}^{N} \left( \frac{S_{t_i} - S_{t_{i-1}}}{S_{t_{i-1}}} \right)^2 \times 100^2,$$

$$RV_{var}^{(2)} = \frac{AF}{N} \sum_{i=1}^{N} \left( \ln \frac{S_{t_i}}{S_{t_{i-1}}} \right)^2 \times 100^2, \tag{12}$$

where $t_i, i = 0, 1, \cdots, N$ is the $i$-th observation time in $[0, T]$, $S_{t_i}$ is the price of underlying asset at the $i$-th observation time $t_i$ and there are altogether $N$ observations. $AF$ is the annualized factor that converts this expression to an annualized variance. Let $\Delta_t = t_i - t_{i-1} = \frac{T}{N}$, the annualized factor $AF = \frac{1}{\Delta_t} = \frac{N}{T}$.

### 3.1 The discrete case

Variance swaps are special kinds of forward contracts. The price of variance swaps are called strike prices in the contracts. In this subsection, we derive the fair strike prices of variance swaps in terms of the joint moment generating function.

We denote $K_{var}^{(i)}, i = 1, 2$ be the fair strike prices of variance swaps with realized variance $RV_{var}^{(i)}, i = 1, 2$. The values of the variance at time $t$ with maturity $T$ can be presented as

$$V_{var}(t) = \mathbb{E}^{\mathbb{Q}}[e^{-r(T-t)}(RV_{var}^{(i)} - K_{var}^{(i)})L|\mathcal{F}_t], \quad i = 1, 2. \tag{13}$$

where $L$ is the nominal amount. To be fair to both parties, the value of the contract should be equal to zero when it is realized. Thus, the fair strike prices $K_{var}^{(i)}, i = 1, 2$ are given by

$$
\begin{aligned}
K_{var}^{(1)} &= \mathbb{E}^{\mathbb{Q}}[RV_{var}^{(1)}|\mathcal{F}_0] = \frac{AF}{N} \sum_{i=1}^{N} \mathbb{E}^{\mathbb{Q}}\left[ \left( \frac{S_{t_i} - S_{t_{i-1}}}{S_{t_{i-1}}} \right)^2 \bigg| \mathcal{F}_0 \right] \times 100^2, \\
K_{var}^{(2)} &= \mathbb{E}^{\mathbb{Q}}[RV_{var}^{(2)}|\mathcal{F}_0] = \frac{AF}{N} \sum_{i=1}^{N} \mathbb{E}^{\mathbb{Q}}\left[ \left( \ln \frac{S_{t_i}}{S_{t_{i-1}}} \right)^2 \bigg| \mathcal{F}_0 \right] \times 100^2.
\end{aligned}
\tag{14}
$$

Since the expectation is additive, we use the additivity of expectations to calculate the fair delivery prices of variance swaps. Specifically, we first calculate the values of $\mathbb{E}^{\mathbb{Q}}\left[ \left( \frac{S_{t_i} - S_{t_{i-1}}}{S_{t_{i-1}}} \right)^2 \big| \mathcal{F}_0 \right]$ and $\mathbb{E}^{\mathbb{Q}}\left[ \left( \ln \frac{S_{t_i}}{S_{t_{i-1}}} \right)^2 \big| \mathcal{F}_0 \right]$. Upon the introduction of variance swaps, we will illustrate the utilization of the joint moment-generating function to compute each term in the aforementioned summation. Consequently, the comprehensive pricing formulas for variance swaps with discrete sampling within the framework (1) are derived in the subsequent sections.

**Theorem 1.** *Let* $K_{var}^{(i)}, i = 1, 2$ *be the fair delivery prices of discretely sampled variance swaps under the dynamics specified in* (1), *Then,*

$$
\begin{aligned}
K_{var}^{(1)} = \frac{100^2 AF}{N} \sum_{i=1}^{N} \Big[ &e^{B(q_2, t_{i-1},)v_0 + C(q_2, t_{i-1})\theta_0 + D(q_2, t_{i-1}) + G(q_2, t_{i-1})} \\
&- 2e^{B(q_4, t_{i-1},)v_0 + C(q_4, t_{i-1})\theta_0 + D(q_4, t_{i-1}) + G(q_4, t_{i-1})} + 1 \Big],
\end{aligned}
\tag{15}
$$

$$
K_{var}^{(2)} = \frac{100^2 AF}{N} \sum_{i=1}^{N} \frac{\partial^2}{\partial \phi^2} e^{B(q_6; t_{i-1})v_0 + C(q_6, t_{i-1})\theta_0 + D(q_6, t_{i-1}) + G(q_6, t_{i-1})} \bigg|_{\phi=0},
\tag{16}
$$

*where*

$$
\begin{cases}
q_1 = (2\ 0\ 0\ 0)^\top, \\
q_2 = (0, B(q_1, \tau), C(q_1, \tau), D(q_1, \tau) + G(q_1, \tau))^\top, \\
q_3 = (1\ 0\ 0\ 0)^\top, \\
q_4 = (0, B(q_3, \tau), C(q_3, \tau), D(q_3, \tau) + G(q_3, \tau))^\top, \\
q_5 = (\phi\ 0\ 0\ 0)^\top, \\
q_6 = (0, B(q_4, \tau), C(q_4, \tau), D(q_4, \tau) + G(q_4, \tau))^\top.
\end{cases}
\tag{17}
$$

*Proof*: (i) Actual rate of return. Since

$$K_{var}^{(1)} = \frac{AF}{N} \sum_{i=1}^{N} \mathbb{E}^{\mathbb{Q}}\left[ \left( \frac{S_{t_i} - S_{t_{i-1}}}{S_{t_{i-1}}} \right)^2 \bigg| \mathcal{F}_0 \right] \times 100^2,$$

we only calculate the expression of $\mathbb{E}^{\mathbb{Q}}\left[\left(\frac{S_{t_i}-S_{t_{i-1}}}{S_{t_{i-1}}}\right)^2\middle|\mathcal{F}_{t_{i-1}}\right]$.

$$
\mathbb{E}^{\mathbb{Q}}\left[\left(\frac{S_{t_i}-S_{t_{i-1}}}{S_{t_{i-1}}}\right)^2\middle|\mathcal{F}_0\right]
$$

$$
= \mathbb{E}^{\mathbb{Q}}\left[\left(e^{x_{t_i}-x_{t_{i-1}}}-1\right)^2\middle|\mathcal{F}_0\right]
$$

$$
= \mathbb{E}^{\mathbb{Q}}\left[e^{2(x_{t_i}-x_{t_{i-1}})}\middle|\mathcal{F}_0\right] - 2\mathbb{E}^{\mathbb{Q}}\left[e^{(x_{t_i}-x_{t_{i-1}})}\middle|\mathcal{F}_0\right] + 1
$$

$$
= \mathbb{E}^{\mathbb{Q}}\left[\mathbb{E}^{\mathbb{Q}}\left[e^{2x_{t_i}}\middle|\mathcal{F}_{t_{i-1}}\right]e^{-2x_{t_{i-1}}}\middle|\mathcal{F}_0\right]
$$

$$
- 2\mathbb{E}^{\mathbb{Q}}\left[\mathbb{E}^{\mathbb{Q}}\left[e^{x_{t_i}}\middle|\mathcal{F}_{t_{i-1}}\right]e^{-x_{t_{i-1}}}\middle|\mathcal{F}_0\right] + 1
$$

$$
= \mathbb{E}^{\mathbb{Q}}\left[e^{B(q_1,\tau)v_{i-1}+C(q_1,\tau)\theta_{i-1}+D(q_1,\tau)+E(q_1,\tau)}\middle|\mathcal{F}_0\right]
$$

$$
- 2\mathbb{E}^{\mathbb{Q}}\left[e^{B(q_3,\tau)v_{i-1}+C(q_3,\tau)\theta_{i-1}+D(q_3,\tau)+E(q_3,\tau)}\middle|\mathcal{F}_0\right] + 1
$$

$$
= e^{B(q_2,t_{i-1},)v_0+C(q_2,t_{i-1})\theta_0+D(q_2,t_{i-1})+G(q_2,t_{i-1})}
$$

$$
- 2e^{B(q_4,t_{i-1},)v_0+C(q_4,t_{i-1})\theta_0+D(q_4,t_{i-1})+G(q_4,t_{i-1})} + 1,
$$

where $q_1$, $q_2$, $q_3$ and $q_4$ are given by Eq (17). Then the delivery price of discretely sampled variance swap is provided as

$$
K_{var} = \frac{100^2 AF}{N}\sum_{i=1}^{N}\left[e^{B(q_2,t_{i-1},)v_0+C(q_2,t_{i-1})\theta_0+D(q_2,t_{i-1})+G(q_2,t_{i-1})}\right.
$$

$$
\left. -2e^{B(q_4,t_{i-1},)v_0+C(q_4,t_{i-1})\theta_0+D(q_4,t_{i-1})+G(q_4,t_{i-1})} + 1\right].
$$

(ii) Log rate of return. Similar to $K_{var}^{(1)}$, we calculate the expression of $\mathbb{E}^{\mathbb{Q}}\left[\left(\ln\frac{S_{t_i}}{S_{t_{i-1}}}\right)^2\middle|\mathcal{F}_0\right]$.

$$
\mathbb{E}^{\mathbb{Q}}\left[\left(\ln\frac{S_{t_i}}{S_{t_{i-1}}}\right)^2\middle|\mathcal{F}_0\right]
$$

$$
= \mathbb{E}^{\mathbb{Q}}\left[(x_{t_i}-x_{t_{i-1}})^2\middle|\mathcal{F}_0\right]
$$

$$
= \mathbb{E}^{\mathbb{Q}}\left[\frac{\partial^2}{\partial\phi^2}e^{\phi(x_{t_i}-x_{t_{i-1}})}\middle|\mathcal{F}_0\right]\Bigg|_{\phi=0}
$$

$$
= \frac{\partial^2}{\partial\phi^2}\mathbb{E}^{\mathbb{Q}}\left[\mathbb{E}^{\mathbb{Q}}\left[e^{\phi x_{t_i}}\middle|\mathcal{F}_{t_{i-1}}\right]e^{-\phi x_{t_{i-1}}}\middle|\mathcal{F}_0\right]\Bigg|_{\phi=0}
$$

$$
= \frac{\partial^2}{\partial\phi^2}\mathbb{E}^{\mathbb{Q}}\left[e^{B(q_5,\tau)v_{i-1}+C(q_5,\tau)\theta_{i-1}+D(q_5,\tau)+E(q_5,\tau)}\middle|\mathcal{F}_0\right]\Bigg|_{\phi=0}
$$

$$
= \frac{\partial^2}{\partial\phi^2}e^{B(q_6;t_{i-1})v_0+C(q_6,t_{i-1})\theta_0+D(q_6,t_{i-1})+G(q_6,t_{i-1})}\Bigg|_{\phi=0},
$$

with $q_5$ and $q_6$ given by Eq (17). As a result, the delivery price of discretely sampled variance swap is presented as

$$
\frac{100^2 AF}{N}\sum_{i=1}^{N}\frac{\partial^2}{\partial\phi^2}e^{B(q_6;t_{i-1})v_0+C(q_6,t_{i-1})\theta_0+D(q_6,t_{i-1})+G(q_6,t_{i-1})}\Bigg|_{\phi=0}.
$$

The proof is completed. □

The pricing formulas for the fair delivery prices of variance swaps under discrete sampling, as determined by Eq (15), are predicated on finite observations. The following subsection investigates the limiting properties of these pricing formulas as $N \to +\infty$ (i.e., $\Delta t \to 0$).

### 3.2 The continuous case

In the previous section, we obtained the fair strike prices of variance swaps under discrete sampling. In this subsection, we will present the pricing formulas for variance swaps under continuous sampling.

The continuous model is widely used for variance swaps. According to [26], the realized variance consists of two parts: The first component accounts for the variance accumulated via the diffusion term in the asset price process, while the second component arises from the jumps in the asset price process. Therefore, the realized variance can be expressed as

$$
\begin{aligned}
RV_{cvar} &= \lim_{N \to +\infty} \frac{100^2}{T} \sum_{i=1}^{N} \left( \frac{S_{t_i} - S_{t_{i-1}}}{S_{t_{i-1}}} \right)^2 \\
&= \frac{100^2}{T} \left[ \int_0^T v_t dt + \sum_{i=1}^{N_T} (e^{J^S} - 1)^2 \right].
\end{aligned}
\tag{18}
$$

As a result, the fair strike price $K_{cvar}$ of the continuously-sampled variance swap can be presented as

$$
\begin{aligned}
K_{cvar} &= \mathbb{E}^{\mathbb{Q}}[RV_{cvar}|\mathcal{F}_0] \\
&= \frac{100^2}{T} \left[ \int_0^T \mathbb{E}^{\mathbb{Q}}[v_t|\mathcal{F}_0] dt + \mathbb{E}^{\mathbb{Q}} \left[ \sum_{i=1}^{N_T} (e^{J^S} - 1)^2 \right] \right].
\end{aligned}
\tag{19}
$$

**Theorem 2.** *The fair strike price $K_{cvar}$ of the continuously-sampled variance swap within the Heston model incorporating stochastic long-term variance mean and jumps can be determined as*

$$
\begin{aligned}
K_{cvar} = \frac{100^2}{T} &\left[ \left( \tilde{\theta} + \frac{\lambda \eta}{\kappa_v} \right) T + \left( v_0 - \tilde{\theta} - \frac{\kappa_v (\theta_0 - \tilde{\theta})}{\kappa_v - \kappa_\theta} \right) \frac{1 - e^{-\kappa_v T}}{\kappa_v} \right. \\
&\left. + \frac{\kappa_v (\theta_0 - \tilde{\theta})}{\kappa_v - \kappa_\theta} \frac{1 - e^{-\kappa_\theta T}}{\kappa_\theta} + \lambda T \left( \frac{e^{2v + 2\delta^2}}{1 - 2\rho_J \eta} - \frac{2 e^{v + \delta^2/2}}{1 - \rho_J \eta} + 1 \right) \right].
\end{aligned}
\tag{20}
$$

*Proof*: We know that $v_t$ and $\theta_t$ satisfy

$$
\begin{cases}
dv_t = \kappa_v (\theta_t - v_t) dt + \sigma_v \sqrt{v_t} dd W_t^v + J^v dN_t, \\
d\theta_t = \kappa_\theta (\tilde{\theta} - \theta_t) dt + \sigma_\theta dW_t^\theta,
\end{cases}
\tag{21}
$$

where $W_t^v$ and $W_t^\theta$ are two independent standard Brownian motions. Integrating both sides of Eq (22), we have

$$
v_t - v_0 = \kappa_v \int_0^t (\theta_s - v_s) ds + \sigma_v \int_0^t \sqrt{v_t} dW_s^v + \sum_{i=1}^{N_T} J_v,
\tag{22}
$$

$$
\theta_t - \theta_0 = \kappa_\theta \int_0^t (\tilde{\theta} - \theta_s) ds + \sigma_\theta \int_0^t dW_s^\theta.
\tag{23}
$$

Taking the expectation on both sides of Eq (23) and Eq (24), and then take the derivative with respect to $t$, one has

$$\frac{d\mathbb{E}^{\mathbb{Q}}[v_t|\mathcal{F}_0]}{dt} = \kappa_v(\mathbb{E}^{\mathbb{Q}}[\theta_t|\mathcal{F}_0] - \mathbb{E}^{\mathbb{Q}}[v_t|\mathcal{F}_0]) + \lambda\eta,$$

$$\frac{d\mathbb{E}^{\mathbb{Q}}[\theta_t|\mathcal{F}_0]}{dt} = \kappa_\theta(\tilde{\theta} - \mathbb{E}^{\mathbb{Q}}[\theta_t|\mathcal{F}_0]),$$

with initial conditions $\mathbb{E}^{\mathbb{Q}}[v_t]|_{t=0} = v_0$ and $\mathbb{E}^{\mathbb{Q}}[\theta_t]|_{t=0} = \theta_0$. By solving the above ODEs, $\mathbb{E}^{\mathbb{Q}}[v_t]$ and $\mathbb{E}^{\mathbb{Q}}[\theta_t]$ can be derived as

$$\mathbb{E}^{\mathbb{Q}}[v_t|\mathcal{F}_0] = \tilde{\theta}(1 - e^{-\kappa_v t}) + \frac{\kappa_v(\theta_0 - \tilde{\theta})}{\kappa_v - \kappa_\theta}(e^{-\kappa_\theta t} - e^{-\kappa_v t}) + v_0 e^{-\kappa_v t} + \frac{\lambda\eta}{\kappa_v}(1 - e^{-\kappa_v t}),$$

$$\mathbb{E}^{\mathbb{Q}}[\theta_t|\mathcal{F}_0] = \tilde{\theta} + (\theta_0 - \tilde{\theta})e^{-\kappa_\theta t}. \tag{24}$$

Since $J^S$ is independent of $N_t$, we obtain

$$\mathbb{E}^{\mathbb{Q}}\left[\sum_{i=1}^{N_T}(e^{J^S} - 1)^2\right] = \lambda T \mathbb{E}^{\mathbb{Q}}\left[(e^{J^S} - 1)^2\right]$$

$$= \lambda T\left(\frac{e^{2v+2\delta^2}}{1 - 2\rho_J\eta} - \frac{2e^{v+\delta^2/2}}{1 - \rho_J\eta} + 1\right). \tag{25}$$

By substituting Eq 24 and Eq 25 into Eq 19 can be readily obtained, which complete the proof.  □

## 4 Numerical examples

We present several numerical examples and compare the obtained conclusions with existing research findings. In our numerical examples, we employ two types of samples, labeled as Sample 1 and 2, where the model parameters are detailed in Table 1.

EXAMPLE 4.1. *In this numerical experiment, we conduct a numerical comparison between the results derived from the pricing formulas and those obtained through Monte Carlo (MC) simulations. Given the similarity in pricing formulas for variance swaps based on actual returns and those based on logarithmic returns, we solely compare the prices of variance swaps based on actual returns with the outcomes of the Monte Carlo simulations. The Euler-Maruyama discretization for the Heston jump-diffusion model with a stochastic long-term jump*

**Table 1. Sample parameters of our model**

| Parameters | $v_0$ | $\theta_0$ | $\kappa_v$ | $\sigma_v$ | $\kappa_\theta$ | $\tilde{\theta}$ | $\sigma_\theta$ | $\rho_{sv}$ |
|---|---|---|---|---|---|---|---|---|
| Sample 1 | 0.1 | 0.15 | 5 | 0.005 | 4 | 0.1 | 0.01 | -0.5 |
| Sample 2 | 0.16 | 0.11 | 6.3 | 0.012 | 3.6 | 0.125 | 0.004 | -0.7 |

Notes: The parameters $r$, $T$, $l$, $\lambda$, $\eta$, $v$, $\rho_J$ and $\delta$ are fixed at $r = 0.01$, $T = 1$, $l = 15$, $\lambda = 0.2$, $\eta = 0.05$, $v = 0.01$, $\rho_J = -0.086$ and $\delta = 0.02$, respectively.

*mean can be expressed as follows*

$$
\begin{cases}
S_{t_i} &= S_{t_{i-1}} e^{(r - \lambda m - \frac{1}{2} v_{t_{i-1}})\Delta t + \sqrt{v_{t_{i-1}}}\sqrt{\Delta t}W_{1,t_{i-1}} + \sum_{N_{t_{i-1}}}^{N_{t_i}} J^S}, \\
v_{t_i} &= v_{t_{i-1}} + \kappa_v(\theta_{t_{i-1}} - v_{t_{i-1}})\Delta t + \sum_{N_{t_{i-1}}}^{N_{t_i}} J^v \\
&\quad + \sigma_v \sqrt{v_{t_{i-1}}}\sqrt{\Delta t}(\rho_{sv} W_{1,t_{i-1}} + \sqrt{1 - \rho_{sv}^2}W_{2,t_{i-1}}), \\
\theta_{t_i} &= \theta_{t_{i-1}}(\tilde{\theta} - \theta_{t_{i-1}})\Delta t + \sigma_\theta \sqrt{\Delta t}W_{3,t_{i-1}},
\end{cases}
\tag{26}
$$

*where $W_{1,t_{i-1}}$, $W_{2,t_{i-1}}$, $W_{3,t_{i-1}}$ are three independent standard normal random variables. $\Delta t = T/N$. $J^S$ and $J^v$ are random variables generated by normal distribution and exponential distribution, respectively.*

 *We exhibit a comparison between the numerical implementation of Eq (15) and the Monte Carlo (MC) simulation results in S1 Fig and Table 2. S1 Fig displays three sets of data: one set represents the variance swap prices derived from 500,000 Monte Carlo simulations based on Eq (14), another set is calculated using Eq (15) for discretely-sampled variance swaps, and the third set is obtained from Eq (20) for continuously-sampled variance swaps. In Table 2, we present a detailed listing of the numerical results for variance swaps under four different sampling frequencies along with their corresponding relative errors. In this table, AS denotes the discretely-sampled analytic solution, MC represents the Monte Carlo simulation results, CON stands for the continuously-sampled analytic solution, and RD signifies the standard error. Additionally, we have recorded the CPU runtime to facilitate a comprehensive comparison of the computational efficiency of the two methods. All calculations were performed on an Intel(R) Core(TM) i5-12500H processor. It is evident from the table that the analytic formula can produce highly accurate results within an extremely short period of time. Notably, as the sampling frequency increases, the time required to compute the fair strike price of variance swaps using Monte Carlo simulation becomes significantly longer than that required using our proposed analytic pricing formula. From S1 Fig and Table 2, we observe that as the sampling frequency increases, the strike price of the variance swap under discrete sampling gradually converges to that under continuous sampling. Furthermore, with increasing sampling frequency, the strike price of the variance swap under discrete sampling becomes very close to the price obtained through MC simulations, this indicates that the results obtained from our analytical results in Theorem 1 are in high agreement with those from the MC simulations.*

 **EXAMPLE 4.2.** *By assigning different values to the parameters in system (1), our model encompasses the Heston model, the Heston model with simultaneous jumps, and the Heston model with stochastic long-run mean of variance. For example, if the parameters $\kappa_\theta = \sigma_\theta = \lambda = 0$, our model deduce to the model framework in [27]. Using the parameters provided by [27], Table 3 presents the analytical solutions derived in this paper, the analytical solutions provided in [27], and the results obtained from Monte Carlo simulations. Clearly, Table 3 demonstrates that our pricing formula is consistent with the closed-form variance swap formula proposed by [27] using the single-factor Heston model. When the parameters satisfy $\kappa_\theta = \sigma_\theta = 0$, our model reduces to the Heston model with simultaneous jumps, the system of equations obtained during the derivation of the joint moment generating function is consistent with [20]. Through computation, we obtain the same pricing formula as in [20] under this condition. On the other hand, when $\lambda = 0$, our model simplifies to the Heston model with a stochastic long-term variance mean proposed by [19], Table 4 presents the analytical solutions derived in this paper, the analytical solutions provided in [19], and the results obtained from Monte Carlo simulations.*

**Table 2. Fair strike prices $K_{var}$ derived from MC simulation, the analytic formula and the continuous model, along with their corresponding relative errors.**

| Sample | N | AS | textbfMC | CON | RD |
|---|---|---|---|---|---|
| Sample 1 | Monthly ($N = 12$) | 1141.0764 | 1145.0146 | 1133.8657 | 0.3439 |
| | CPU time (s) | 0.0115 | 1.1463 | - | - |
| | Fortnightly ($N = 26$) | 1137.1822 | 1138.5908 | 1133.8657 | 0.1237 |
| | CPU time (s) | 0.0141 | 1.6675 | - | - |
| | Weekly ($N = 52$) | 1135.5212 | 1136.1883 | 1133.8657 | 0.0587 |
| | CPU time (s) | 0.01185 | 2.7977 | - | - |
| | Daily ($N = 252$) | 1134.2058 | 1134.3083 | 1133.8657 | 0.0090 |
| | CPU time (s) | 0.0147 | 11.4784 | - | - |
| Sample 2 | Monthly ($N = 12$) | 1288.9977 | 1288.5884 | 1280.6165 | -0.0318 |
| | CPU time (s) | 0.0071 | 1.1323 | - | - |
| | Fortnightly ($N = 26$) | 1284.4478 | 1283.9812 | 1280.6165 | -0.0363 |
| | CPU time (s) | 0.0087 | 1.8132 | - | - |
| | Weekly ($N = 52$) | 1282.5236 | 1282.3325 | 1280.6165 | -0.0149 |
| | CPU time (s) | 0.0092 | 3.6321 | - | - |
| | Daily ($N = 252$) | 1281.0086 | 1280.8717 | 1280.6165 | -0.0107 |
| | CPU time (s) | 0.0136 | 10.6012 | - | - |

Notes: We provide numerical results for variance swaps under four different sampling frequencies, along with their corresponding relative errors. In this table, AS, MC, CON and RD represent the discretely-sampled analytic solution, Monte Carlo simulation results, continuously-sampled analytic solution and standard error, respectively.

**Table 3. Results of Heston model using parameters in [27]**

| Sampling frequency | MC | Zhu's formula | Our formula |
|---|---|---|---|
| Monthly ($N = 12$) | 243.2 | 242.7 | 242.71 |
| Fortnightly ($N = 26$) | 238.1 | 238.6 | 238.58 |
| Weekly ($N = 52$) | 237.4 | 237.1 | 237.11 |
| Daily ($N = 252$) | 236.5 | 236.1 | 236.08 |

Notes: The table presents the results of the Heston model using different sampling frequencies. The values are compared between MC simulation, Zhu's formula, and our proposed formula.

**Table 4. Results of our model using parameters in [19]**

| Sample | N | Our formula | Yoon's formula | MC |
|---|---|---|---|---|
| Sample1 | Monthly ($N=12$) | 953.168 | 953.145 | 948.340 |
| | Weekly ($N=52$) | 950.843 | 950.838 | 948.411 |
| | Daily ($N=252$) | 950.248 | 950.248 | 951.106 |
| Sample2 | Monthly ($N=12$) | 1272.507 | 1272.510 | 1276.980 |
| | Weekly ($N=52$) | 1267.881 | 1267.880 | 1265.570 |
| | Daily ($N=252$) | 1266.665 | 1266.670 | 1266.740 |
| Sample3 | Monthly ($N=12$) | 775.506 | 775.506 | 773.783 |
| | Weekly ($N=52$) | 773.478 | 773.478 | 772.599 |
| | Daily ($N=252$) | 772.970 | 772.970 | 773.570 |

Notes: The table presents the results of our model using different sampling frequencies and samples. The values are compared between our formula, Yoon's formula, and MC simulations.

**EXAMPLE 4.3.** *We investigated the impact of key parameters $\kappa_\theta$, $\tilde{\theta}$, $\sigma_\theta$, and $\theta_0$ of the stochastic long-term mean on the strike price of variance swaps, while keeping all other parameters in the model consistent with those in Sample 2.*

From *S2 Fig*, we observe that the primary parameters of the stochastic long-term mean significantly impact the strike price of the variance swap. Specifically, the strike price of the variance swap increases with an increase in $\kappa_\theta$. This is because $\kappa_\theta$ determines the speed at which the long-term mean $\theta_t$ reverts to its long-term average $\tilde{\theta}$. Specifically, when $\kappa_\theta$ is high, $\theta_t$ converges to $\tilde{\theta}$ more rapidly. Over the life of the contract, this leads to an increase in the variability of $v_t$, which in turn amplifies the fluctuations in realized returns, thereby raising the fair strike price of the variance swap. Conversely, when $\kappa_\theta$ is low, the reversion of $\theta_t$ to $\tilde{\theta}$ occurs more slowly. Over the contract period, the variability of $v_t$ becomes relatively smaller, resulting in reduced fluctuations in realized returns and, consequently, a lower fair strike price for the variance swap. The strike price of the variance swap exhibits an increasing trend with an increase in $\tilde{\theta}$ and $\theta_0$, and the increase occurs at a relatively steady rate. This is because when $\tilde{\theta}$ is high, $\theta_t$ tends to fluctuate around a higher level, thereby pulling $v_t$ toward higher values. This increases the variability of $\frac{S_{t_i}-S_{t_{i-1}}}{S_{t_i}}$, leading to a higher realized variance and, consequently, a higher fair strike price. In markets with low volatility expectations (e.g., during periods of economic stability or calm market sentiment), investors demand lower risk premiums. As a result, the fair strike price of variance swaps decreases. Additionally, if the market's expectation of future volatility is high at the initial time, the volatility of asset prices will be greater in the early stages of the contract, resulting in higher realized variance and, consequently, an increase in the fair strike price of the variance swap. Furthermore, the strike price of the variance swap increases with an increase in $\sigma_\theta$, and the rate of increase gradually decreases with an increase in the sampling frequency. The increase in $\sigma_\theta$ leads to a higher strike price for variance swaps due to the greater uncertainty and instability it introduces into the long-term volatility dynamics. This uncertainty increases the risk premium demanded by market participants, driving up the price of variance swaps. However, as sampling frequency increases, the rate at which the strike price rises slows down, reflecting the averaging effect of more frequent measurements.

**EXAMPLE 4.4.** *We further investigated the impact of key jump parameters $\lambda$, $\eta$, $v$, and $\delta$ on the strike price of variance swaps, while keeping all other parameter settings in the model consistent with those in Sample 2.*

Observing *S3 Fig* reveals that the key parameters related to the jump component play a crucial role in determining the strike price of the variance swap. Specifically, the variance swap shows a high sensitivity to $\lambda$, with the strike price gradually increasing as $\lambda$ increases. Specifically, the jump intensity $\lambda$ reflects the probability of price jumps in the underlying asset within a unit of time. A higher jump intensity implies a greater likelihood of significant short-term fluctuations in the underlying asset price, indicating an expected increase in future volatility, leading to an increase in the price of the variance swap. A higher $\lambda$ increases the uncertainty about future asset price movements. Investors and hedgers demand greater compensation for this heightened uncertainty, thereby leading to an increase in the fair strike price of the variance swap. The strike price of the variance swap gradually increases with an increase in $\eta$. This is because an increase in $\eta$ will increase the volatility of the variance, leading to an increase in the volatility of the underlying asset price, thereby increasing the strike price of the variance swap. This increased variability in volatility translates to higher uncertainty about future market conditions. As a result, the strike price of the variance swap rises to reflect the heightened risk. The strike price of the variance swap shows an increasing trend with an increase in $v$ and $\delta$, respectively, and the rate of price increase gradually accelerates. This is because an increase in $v$ and $\delta$ will lead to more pronounced market price movements, i.e., an increase in market volatility, resulting in an increase in the strike price of the variance swap. An increase in $v$ and $\delta$ signifies an elevated probability of extreme price fluctuations. On the one hand, this augments the risk premium demanded by investors and hedgers, as they seek compensation for the heightened uncertainty. On the other

*hand, in markets characterized by high levels of $v$ and $\delta$, variance swaps become more valuable as hedging instruments, thereby elevating their strike prices. This may be attributed to the heightened demand for effective volatility protection stemming from the possibility of large and unpredictable price jumps, making variance swaps a pivotal tool for risk management in such environments.*

## 5 Conclusions

This paper focuses on pricing variance swaps within the framework of the Heston jump-diffusion model with a stochastic long-term mean. We first derive the joint moment-generating function, which is then used to derive the pricing formula for variance swaps with discrete sampling. Subsequently, to further investigate the limiting properties of the pricing formulas under discrete sampling, we also derive the pricing formula for variance swaps with continuous sampling. A series of numerical experiments are conducted to validate the accuracy and reliability of the pricing formula for discrete sampling. The numerical results computed using the discrete sampling formula are compared with Monte Carlo simulations and previous research findings. The results demonstrate a close alignment between our discrete sampling pricing formula, Monte Carlo simulation results, and existing research, confirming its effectiveness. Additionally, a thorough analysis is conducted on the specific impact of key parameters (such as $\sigma_\theta$ and $\lambda$) related to the stochastic long-term mean and jump risk on the prices of variance swaps. The research findings indicate that these parameters significantly influence the prices of variance swaps, further emphasizing the importance of considering stochastic long-term mean and jump risk in pricing variance swaps.

## Supporting information

**S1 Fig. The comparison of prices of variance swap obtained from our formulas and obtained from MC simulations.**
(EPS)

**S2 Fig. The impact of the key parameters of the stochastic long-term mean on the strike price of variance swaps under discrete sampling.**
(EPS)

**S3 Fig. The impact of the key parameters of jump risk on the strike price of variance swaps under discrete sampling.**
(EPS)

**S1 Dataset. The dataset associated with S1 Fig, generated through numerical simulations conducted using MATLAB.**
(PDF)

**S2 Dataset. The dataset associated with S2 Fig, generated through numerical simulations conducted using MATLAB.**
(PDF)

**S3 Dataset. The dataset associated with S3 Fig, generated through numerical simulations conducted using MATLAB.**
(PDF)

## Author contributions

**Conceptualization:** Jing Fu.

**Data curation:** Jing Fu.

**Formal analysis:** Jing Fu.

**Funding acquisition:** Jing Fu.

**Investigation:** Jing Fu.

**Methodology:** Jing Fu.

**Project administration:** Jing Fu.

**Resources:** Jing Fu.

**Software:** Jing Fu.

**Supervision:** Jing Fu.

**Validation:** Jing Fu.

**Visualization:** Jing Fu.

**Writing – original draft:** Jing Fu.

**Writing – review & editing:** Jing Fu.

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
