## [Decision Letter · Decision Letter 0]

3 Jan 2025

PONE-D-24-55815Analytic solutions of variance swaps for Heston models with  stochastic long-run mean of variance and jumpsPLOS ONE

Dear Dr. Fu,

Thank you for submitting your manuscript to PLOS ONE. After careful consideration, we feel that it has merit but does not fully meet PLOS ONE’s publication criteria as it currently stands. Therefore, we invite you to submit a revised version of the manuscript that addresses the points raised during the review process.

We look forward to receiving your revised manuscript.

Kind regards,

Mazyar Ghadiri Nejad, Ph.D.

Academic Editor

PLOS ONE

Journal Requirements:

3. We note that your Data Availability Statement is currently as follows: “All relevant data are within the manuscript and in Supporting Information files.”

Reviewers' comments:

Reviewer's Responses to Questions

**Comments to the Author**

1. Is the manuscript technically sound, and do the data support the conclusions?

Reviewer #1: Yes

Reviewer #2: Yes

2. Has the statistical analysis been performed appropriately and rigorously? 

Reviewer #1: Yes

Reviewer #2: Yes

3. Have the authors made all data underlying the findings in their manuscript fully available?

Reviewer #1: Yes

Reviewer #2: Yes

4. Is the manuscript presented in an intelligible fashion and written in standard English?

Reviewer #1: Yes

Reviewer #2: Yes

5. Review Comments to the Author

Reviewer #1: Summary and overall impression:

The aim of the paper is to present the pricing formulas for variance swaps within the Heston model that incorporates jumps and stochastic long-term mean for the underlying asset. The author used partial -integro-differential equations to obtain the joint moment-generating function for the proposed model. Moreover, the proposed parameters of the model makes it more comprehensive than the existing models in literature. An experiment on the impact of parameter variation on the strike price of variance swaps is done through a Mont Carlo (MC) simulation and it showed consistency in the results obtained. Furthermore, the author confirmed through several numerical examples the accuracy of the proposed model by comparing its results with the results of existing models in the literature.

Suggestion

I suggest the author uses Origin or OriginPro for scientific graphing to make the presented graphs more clear.

Reviewer #2: The paper presents an innovative extension of the Heston model by incorporating a stochastic long-term mean and jump risk, providing rigorous analytical pricing formulas for variance swaps under both discrete and continuous sampling. The methodology is well-supported by numerical experiments, including Monte Carlo validations, and offers valuable insights into the impact of key parameters on pricing. While minor improvements, such as expanded sensitivity analyses and discussions on computational efficiency, are suggested, the paper significantly contributes to the field of financial derivatives pricing. I recommend its acceptance after minor revisions to enhance clarity and practical applicability.

6. PLOS authors have the option to publish the peer review history of their article (what does this mean?). If published, this will include your full peer review and any attached files.

Reviewer #1: No

Reviewer #2: No

---

## [Author Response · Author response to Decision Letter 1]

13 Jan 2025

Dear Editor,

Thank you for your email and for the constructive feedback provided by the reviewers. We greatly appreciate the opportunity to revise our manuscript and address the points raised during the review process.

We have carefully considered the reviewers’ comments and have made the necessary revisions to the manuscript. Below, we outline the specific changes made in response to each reviewer’s suggestions:

Reviewer #1’s Suggestion on Graph Clarity:

We have re-plotted the graphs using Matlab and uploaded the images in EPS format to improve their clarity and presentation quality.

Reviewer #2’s Suggestions:

We have expanded the sensitivity analysis to provide a more comprehensive understanding of the impact of key parameters on pricing.

We have included additional discussions on the computational efficiency of the proposed model to further highlight its practical applicability.

Additionally, we have ensured that our manuscript complies with PLOS ONE’s formatting requirements and have uploaded the data to confirm that all raw data required to replicate the results of our study are included in the manuscript and Supporting Information files.

Attached to this email, please find the following documents:

A rebuttal letter detailing our responses to each point raised by the reviewers.

A marked-up copy of the manuscript highlighting the changes made.

An unmarked version of the revised manuscript.

We hope that the revised manuscript meets the journal’s publication criteria. Should any further revisions be required, please do not hesitate to contact us.

Once again, thank you for your time and consideration.

Sincerely,

Dr. Fu

School of Mathematics

Southwestern University of Finance and Economics

Chengdu, Sichuan, China, 611130

---

## [Decision Letter · Decision Letter 1]

24 Jan 2025

Analytic solutions of variance swaps for Heston models with  stochastic long-run mean of variance and jumps

PONE-D-24-55815R1

Dear Dr. Jıng Fu,

We’re pleased to inform you that your manuscript has been judged scientifically suitable for publication and will be formally accepted for publication once it meets all outstanding technical requirements.

Kind regards,

Mazyar Ghadiri Nejad, Ph.D.

Academic Editor

PLOS ONE

Reviewers' comments:

Reviewer's Responses to Questions

**Comments to the Author**

1. If the authors have adequately addressed your comments raised in a previous round of review and you feel that this manuscript is now acceptable for publication, you may indicate that here to bypass the “Comments to the Author” section, enter your conflict of interest statement in the “Confidential to Editor” section, and submit your "Accept" recommendation.

Reviewer #2: All comments have been addressed

2. Is the manuscript technically sound, and do the data support the conclusions?

Reviewer #2: Yes

3. Has the statistical analysis been performed appropriately and rigorously? 

Reviewer #2: Yes

4. Have the authors made all data underlying the findings in their manuscript fully available?

Reviewer #2: Yes

5. Is the manuscript presented in an intelligible fashion and written in standard English?

Reviewer #2: Yes

6. Review Comments to the Author

Reviewer #2: The authors have done an excellent job with the revised manuscript. Well done! My decision is to accept it. Congratulations!

7. PLOS authors have the option to publish the peer review history of their article (what does this mean?). If published, this will include your full peer review and any attached files.

Reviewer #2: No

---

## [Editor Report · Acceptance letter]

PONE-D-24-55815R1

PLOS ONE

Dear Dr. Fu,

I'm pleased to inform you that your manuscript has been deemed suitable for publication in PLOS ONE. Congratulations! Your manuscript is now being handed over to our production team.

Kind regards,

on behalf of

Assoc. Prof. Dr. Mazyar Ghadiri Nejad

Academic Editor

PLOS ONE